# Technology-Based Motivation Support for Seniors’ Physical Activity—A Qualitative Study on Seniors’ and Health Care Professionals’ Views

**DOI:** 10.3390/ijerph16132418

**Published:** 2019-07-08

**Authors:** Maria Ehn, Ann-Christin Johansson, Åsa Revenäs

**Affiliations:** 1School of Innovation, Design and Engineering, Mälardalen University, SE-721 23 Västerås, Sweden; 2School of Health, Care and Social Welfare, Mälardalen University, SE-721 23 Västerås, Sweden

**Keywords:** physical activity, senior, technical support, motivation, behavioral change techniques (BCTs), self-determination theory (SDT), unified theory of acceptance and use of technology (UTAUT)

## Abstract

This paper investigates seniors’ and health care professionals’ (HCPs) perceptions on needed contributions and qualities of digital technology-based motivation support for seniors’ physical activity (PA). Seniors and HCPs expressed their views in focus groups, which were analyzed separately by inductive content analysis. Similarities and differences in seniors’ and HCPs’ views were identified through thematic analysis of qualitative results from both focus groups. This article’s main findings are that both seniors and HCPs believed digital technology should support and make PA more enjoyable in ways to strengthen seniors’ control and well-being. However, seniors emphasized support for social interaction, while HCPs also requested support for increasing seniors’ insight into PA and for facilitating their dialogue with seniors. Conclusions to be drawn are that seniors and HPCs shared overall views on digital technology’s main contributions but had different perspectives on how those contributions could be obtained. This highlights the importance of the early identification of user groups and exploration of their different needs when developing new solutions. Moreover, seniors’ and HCPs’ perceptions included aspects relevant for personal motivation, technology acceptance, and PA behavioral change according to self-determination theory, unified theory of acceptance and use of technology, and behavioral change techniques for increasing PA.

## 1. Introduction

Physical activity (PA) is defined as “any bodily movement produced by skeletal muscles that results in energy expenditure” [1]. PA has several health benefits for the population in general [2]. The health benefits of regular PA for older adults are extensive and include cardiovascular, musculoskeletal, and psycho–social health [3]. Moreover, PA can contribute to preventing falls in community-dwelling seniors, especially exercise programs that challenge balance and are of higher dose (at least three hours of exercise per week) [4]. International recommendations on the types and amount of PA needed to improve and maintain health have been formulated [5]. Recommendations for persons aged 65+ are the same as for adults in general with the addition that: (1) persons with poor mobility should perform PA to enhance balance and prevent falls three or more days per week; (2) when older adults cannot do the recommended amounts of PA due to the presence of health conditions, they should be as physically active as their abilities and conditions allow. Examples of activities to enhance balance and to prevent falls include group and home-based exercise programs and tai-chi [6]. Approximately one-third of adults worldwide fail to reach public health guidelines for recommended levels of PA, and the proportion of persons with insufficient PA in relation to guidelines increases with age [7]. Moreover, seniors’ uptake and adherence to fall-prevention PA is low [8,9]. Increased and long-term maintenance of seniors’ PA (in general and related to fall prevention) is, therefore, highly important for public health.

A review of the effectiveness of PA interventions among adults aged 60+ concluded that PA interventions for seniors were more effective when containing cognitive-based and cognitive-behavioral approaches than behavioral-only approaches [10]. Moreover, it has been recommended that interventions for promoting older people’s engagement in PAs to prevent falls should be tailored to the specific values and situation of the individual [11]. In addition, a systematic review and thematic synthesis of qualitative studies on perspectives on PA among people aged 60+ concludes that strategies to enhance seniors’ PA should include raising awareness of the benefits and minimizing the perceived risks of PA as well as improving the environmental and financial access to PA opportunities [12]. Digital technology-based applications can be used to deliver more efficient and effective fall prevention exercise and PA interventions among seniors [13]. For example, the potential of wearable activity monitors for promotion of seniors’ PA has been investigated [14,15,16,17].

Technology-based solutions need to be accepted and used by the intended users. In order to design new digital applications users are interested in and willing to use, user-centered design processes are often used [18]. Here, the users’ needs, requirements, and perceptions are used in the design process. Moreover, new technical applications supporting PA should be designed in accordance with relevant theory [19]. It is, therefore, relevant to relate needs, requirements, and perceptions gathered from users to relevant theories for technology use and for changing and sustaining PA behavior. In particular, the degree to which the users are willing to use technical applications is described by the unified theory of acceptance and use of technology (UTAUT) [20]. According to UTAUT, four constructs (performance expectancy; effort expectancy; social influence and facilitating conditions, respectively) are determinants of users’ intent and usage of a system. Rhodes et al. [19] have given an overview of PA behavior theories and their key constructs. For example, the behavioral change wheel [21] and taxonomy of behavioral change techniques (BCTs) [22,23] provide systematic approaches for designing, characterizing, evaluating, reporting, and implementing behavioral change interventions. The taxonomy is particularly relevant for defining content and evaluating behavioral effects of PA interventions. In addition, autonomous motivation has been proven as a determinant of sustained PA behavior [24,25]. Autonomous motivation is a central construct of the self-determination theory (SDT) [26,27], which comprises five complementary mini-theories and provides a viable framework for explaining motivation quality and PA behavior [19]. A systematic literature review made by Teixeira et al. [28] concluded that there is good evidence for the value of SDT in understanding exercise behavior.

The study reported in this article was conducted as a first step in a user-centered design process [18] of digital technology supporting and motivating seniors to perform PA. The main aim was to investigate seniors’ and health care professionals’ (HCPs) perceptions on possible contributions and qualities needed/required from technology in supporting and motivating seniors to perform PA. The HCPs’ perspective was considered relevant here, since they also could be users of the technology when coaching seniors to perform PA. In order to be able to design and develop a solution that is useful for both user groups, it is important to gain an understanding of the differences and similarities among the two user groups’ views. Hence, seniors and HCPs were essential for the feasibility of the study by representing two different potential user groups of the technology, while the role of the researchers was to investigate the users’ views and perspectives on the technology and use the results in the user-centered process. A secondary aim was to discuss whether the views expressed were in accordance with the SDT [26,27], UTAUT [20] or reflected elements in the BCT taxonomy for PA behavioral change [23]. The following research questions were addressed:

Q1: What are seniors’ and HCPs’ views on needs and requirements on digital technology for supporting and motivating seniors to increase PA?

Q2: Which similarities and differences can be found between seniors’ and HCPs’ views?

Principal conclusions from the study are:

(1) Both seniors and HCPs saw that technology should support PA and make it more enjoyable while strengthening the seniors’ control and well-being. However, the seniors’ opinions were related to social aspects, enjoyment, and how the technology could contribute to making seniors feel better, whereas HCPs highlighted how the technology could contribute to their role as professional coaches and described how the technology could facilitate the dialogue between the senior and coaching HPC as well as providing information. Both seniors’ and HCPs’ views can be supported by SDT. Also, BCTs relevant for increasing seniors’ PA can be identified in the results.

(2) Seniors and HPCs had similar views on qualities that the technology should have in order to be useful and attractive. Views expressed were mainly in accordance with UTAUT.

## 2. Materials and Methods

The study was part of a larger project where a user-centered development of the technology-based motivation support for PA was prepared. The larger project was a collaboration between the university, municipality, county council, and three companies and included a reference group for users (called user forum) including senior organizations and a sports association which also provides specific activities for seniors.

### 2.1. Study Design

This study was explorative with a qualitative design [29], aimed at gaining an understanding of seniors’ and HCPs’ perceptions of possible contributions and qualities needed and required to support and motivate seniors to be physically active. Data on the perceptions of the two groups were collected through focus group interviews. In order to answer the study’s first research question (What are seniors’ and HCPs’ views on needs and requirements on digital technology for supporting and motivating seniors to increase PA?), the content of the focus group interviews with seniors and HCPs was analyzed by inductive qualitative content analysis according to Graneheim and Lundman [30]. Thereafter, the study’s second research question (What similarities and differences can be found between seniors’ and HCPs views?) was addressed in a second step of analysis. Here, sub-categories of the analyses of seniors’ and HCPs’ views were further analyzed according to principles of inductive content analysis in order to identify similarities and differences among the two user groups. An overview of the study design (including research questions, approaches used to address the questions, and results) is presented in Figure 1. The seniors and HCPs represented two different potential user groups of the technology, while the researchers investigated their views and perspectives.

### 2.2. Ethical Considerations

Prior to conducting the study, the researcher responsible for the focus group interviews (A.C.J) investigated whether the planned study required ethical vetting by consulting a member of a regional ethical board committee. The investigation showed that ethical vetting was not required for this type of study, mainly for the following reasons: (1) Participants were recruited and expressed their views as private persons (seniors) or professionals (health care professionals). Hence, no humans participated in the study in the role of patients. (2) No sensitive personal information (according to definition in laws presented below) was collected from the participants. The sole information collected from the participants was their views expressed in the focus groups interviews, which focused on the participants’ views on technology-based support for PA (interview guides are presented in Appendix A and Appendix B). This focus is important since focus group interview is a data collection method in which the integrity of participants must be carefully considered. (3) The methods used in the study did not include physical interventions, nor were they aimed at affecting the research participant. No risks of personal injuries could be identified.

The study was performed according to good research practices including provision of written and oral information to interested persons in the recruitment process, voluntary participation, and secure data management. Moreover, the reasons described above were compared with paragraphs 3–4 of The Act Concerning the Ethical Review of Research Involving Humans (SFS 2003:460) [30] and information on ethical vetting according to the law is presented on the Swedish website “CODEX rules and guidelines for research”, which is hosted by the Swedish Research Council in cooperation with The Centre for Research Ethics and Bioethics at Uppsala University [31].

### 2.3. Recruitement and Participants

Two different purposive samples were recruited: seniors and HCPs.

#### 2.3.1. Seniors

Senior participants, 65 years of age or older, and living independently were recruited from senior organizations and a sports association that also provides specific activities for seniors. All mentioned organizations were participants in the user forum for the larger project this study was included in. The user forum representative of each organization/association informed members in their home organization about the project and the planned focus group interview. Members that were interested in participating in the study announced their interest verbally to the user forum representative. The representative asked for the member’s permission to share his/her contact information with the researchers. The researcher (A.C.J.) then contacted interested persons by telephone calls, gave oral information about the study, explained that the study would be done as a group interview at one occasion, the time planned for the interview, and also practical information on how and when the interview was planned. It was also emphasized that participation was voluntary and in connection to this, seniors had the possibility to raise questions about the study and participation. After this conversation, written information was sent to the senior if he or she still was interested to participate in the study. No senior who had announced interest declined participation after he/she had received information. Written, informed consent was collected from all participants in connection to the focus group interview.

Seven seniors, four males and three females, participated in the focus group interview. The interview started with information on how the focus group interview was planned to be accomplished and then the participants made a brief presentation of themselves, including their personal views on PA. Some of the participants mentioned their age in their personal presentation, the range of ages mentioned was 66–82 years old. Several participants described that they were physically active, and that PA played an important role in their lives. There were also participants with an active lifestyle in general but without special focus on PA. Some of the participants had active roles in promoting seniors’ PA through engagement in senior and sport organizations.

Motivators for PA expressed by the participants included having a strong focus on PA in daily life (established earlier in life through different roles on physical training or own training), setting personal goals for exercise results, wanting to delay ageing, willing to continue with exercises and PAs that one has done for a long time, being able to participate in sport competitions with good results, wanting to feel good/prevent physical pain by being active, and needing to take the dog for a walk. Hinders for PA expressed in the group included feeling tired and having difficulties in getting up after sitting down taking a rest, feeling resistant to leave home in the evening, being lazy sometimes, being sick, injured or physically disabled.

#### 2.3.2. Health Care Professionals

Participants working at health care organizations within the municipality or the county council were recruited. They were recruited through their employers, which were partners in the larger project that included the study. Via contact persons in the project from each organization, the researcher A.C.J. obtained names and contact information of HCPs with relevant expertise and interested in participating in the study. The participants had different professional backgrounds and roles in work related to PA, health, and well-being of seniors. Prior to the study, information was sent via email to the HCPs who were considered suitable for the study requirements. This email contained information about the study, that it would be done as a group interview at one occasion, the time planned for the interview, and practical information on how and when the interview was planned. It was also emphasized that participation was voluntary and the HCPs had the possibility to raise questions about the study and participation. Written, informed consent was collected from all participants in connection to the focus group interview. No HCP who was asked for participation declined participation in the study.

Eight HCPs from the municipality or county council participated in the focus group interview. Six of the professional participants were female and two were male. The participants represented different professional perspectives such as public health and physiotherapy. Also, the participants had different professional roles related to health and PA of senior citizens: several participants worked to promote health and PA of senior citizens in independent living, through counselling and leading activities at senior training facilities. Some of the participants also provided physiotherapy exercises for seniors in special care residences or in rehabilitation of senior patients. There were also participants working at the strategic level in health care organizations and working with welfare technology in special care and home care. The focus group interview was accomplished in an analogous way to how it was done with seniors.

### 2.4. Data Collection

Two different focus group interviews were conducted at the university, one with seniors and one with HCPs. The interviews lasted for 60–90 min. The focus group interviews were led by a discussion moderator (A.C.J) with wide experience as a physiotherapist and researcher, familiar with interviews as both a clinician and researcher. A second researcher (M.E.) also supported the interviews in the role of assessor, taking notes during the discussion as well as posing additional questions and concluding the discussions in the end of the interviews. The participants were preliminary informed that the purpose of the interviews was to explore their perceptions of how technology could support and motivate seniors to be physically active and that a special focus in the discussion would be on their views on needs and requirements that the technology should meet. Semi-structured interview guides were used (Appendix A and Appendix B, respectively). Both group interviews were audio-recorded and transcribed verbatim. In the opening of the focus group interviews, the participants were asked to say something about themselves and their view on PA. In addition, the HCPs were asked to explain how their professional role related to supporting and motivating seniors to increase their PA.

### 2.5. Data Analysis

#### 2.5.1. Analysis of Seniors’ and HCPs’ Views Expressed in Focus Group Interviews

Qualitative data describing the participants were summarized for each focus group. The qualitative content of each focus group interview was analyzed by inductive qualitative analysis [32,33]. In the analysis, categories and subcategories were abstracted from the content of the transcribed interviews. The analysis started with M.E. reading the two transcripts carefully. Consecutively, the texts were read phrase by phrase and meaning units related to needs and requirements on technology supporting and motivating seniors to be physically active were extracted. The extracted meaning units were condensed through discussion and set in agreement by A.C.J. and M.E. The condensed meaning units were analyzed and coded by M.E. and Å.R. Thereafter, the codes were discussed and set in agreement by M.E. and Å.R. In the analysis, the condensed meaning units were printed out on paper slips and were aggregated into codes. Codes were printed manually with pencil on Post-it Notes. The codes were sorted and abstracted into categories, based on similarity in the content, and subcategories reflecting/illustrating different aspects of each category. On a practical level, the abstraction of codes and sub-categories was performed manually by grouping Post-it Notes representing codes and sub-categories on large pieces of paper. The resulting hierarchy of themes/categories/sub-categories/codes was documented in Microsoft Excel to facilitate further discussion and modification of the hierarchy and labels of themes, categories, and sub-categories.

Both M.E. and Å.R. followed each step in the analysis and discussed codes, categories, and subcategories until consensus was reached. The final version of the analysis was read by M.E., Å.R., and A.C.J. to ensure the rigor of the described categories and subcategories [33]. In order to strengthen the trustworthiness of the analysis, quotations from the interviews were used to illustrate and exemplify views behind the described categories, subcategories, and codes.

#### 2.5.2. Analysis of Similarities and Differences in Views Expressed by Seniors and HCPs

The qualitative results of the two focus groups were further analyzed using the principles of inductive qualitative content analysis [32,33]. All sub-categories from the analyses were abstracted according to similarities in content and labeled to reflect this similarity. In the search for similarities, two dimensions emerged: the first dimension contained sub-categories illustrating how the technology was perceived to contribute to increased PA. The sub-categories within this dimension were abstracted according to similarities in content and labeled to reflect this similarity. These new groups of sub-categories were called “sub-contributions” and were further categorized into “identified contributions” (i.e., categories according to Graneheim and Lundman [30]). The other dimension illustrated qualities that the technology (on a system level) should have in order to be useful and attractive for seniors and HCPs. Also, here, similarities were searched for; these evolved in the analysis and were grouped into “identified qualities”. According to the described procedure, M.E. produced a first draft of the structured comparison in Microsoft Excel. This was reviewed by Å.R. who suggested modifications of the structure and re-labelling of certain sub-contributions/qualities and identified contributions/qualities. Finally, M.E. and Å.R. discussed the content and labelling of both dimension and the included contributions and qualities until consensus was reached. The final version of the analysis was read by A.C.J. to ensure the rigor of the analysis [33]. This resulted in further clarifications in the analysis process.

## 3. Results

### 3.1. Focus Group Interviews

#### 3.1.1. Seniors Views on Needs and Requirements on Technology-Based Motivation Support for PA

Two themes emerged in the inductive qualitative analysis. Each theme contained three categories, each including 2–5 adherent sub-categories. Themes, categories, and sub-categories are presented in Appendix C and described below.

Theme 1: A help for the user in daily life

The first theme described how the technical support should help the user in daily life. As can be seen in Appendix C, adherent categories illustrated that the technology must be surmountable, customizable, and helpful in facilitating PA in daily life. Both practical and emotional aspects were described.

##### Surmountable

The participants described that the technical solution needs to be surmountable, both practically and emotionally. On a practical level, the technology needs to be easy to use, operate, and understand. It was also pointed out that devices need to be of a manageable size, not too small and robust, meaning they should both work stably and not be heavy-duty. It was expressed that the support must not be perceived as demanding, and therefore, should work without requiring the user to remember extra things to be done. It was also emphasized that technical applications must allow the user to choose when he/she will perform physical activities.
“That (a step counter) was not so fun. You needed to remember to put it on.”


The seniors stressed that the technology must not provoke fear among the users. Envisioned fear could both include not being able to manage the technology, making a fool of oneself, and having to ask for help. Also, fear of being forced into something unwanted and of ruining something was described.
“I think many people are afraid of technology. These things with credit cards and phones and people calling pretending to be…That strikes back on all technology in a way.”


Finally, the need for access to help was emphasized, especially with installation and when beginning to use the technical applications. However, for further help on how to use the technology, it was envisioned that both younger persons and other seniors can contribute.

##### Customizable

In order to suit the individual in daily life, the participants perceived that the support must be able to integrate in different aspects of daily life. They described that using the technology should easily be integrated in daily routines and saw that the activity monitor needed to be integrated in things already worn. Moreover, they perceived that feedback should be provided on technical platforms already present at home, preferably devices normally used on a daily basis.
“Imagine that I came home one evening and when I sat down in front of the TV, sitting down there is something most people do anyway, and the first thing happening is that the screen asks me How am I today? What do I need to do to feel better?”


The participants also emphasized that the solution should be modular, both in terms of functionality and in terms of provision of feedback. A core module with basic functionality possibly extensible with new modules with additional functionalities was preferred. Being able to choose how extensive the personal feedback should be was considered important since it was envisioned that some persons could be motivated by extensive feedback while this could be overwhelming and tiring for others.

##### Helpful Facilitator

The participants described the importance of really feeling that the technology is supportive, helpful, facilitating, and beneficial. They expressed that they needed to feel that the technology contributes to their well-being and that it is really needed. They also emphasized that the technical support should facilitate the performance of PA in daily life and wanted the technical applications to summarize PA that had been performed over a day.
“But it is all the short distances you walk, if you summarize them every day. But that really adds up to something. Then you are adding on to the sum. That sounded a bit exciting”.


Theme 2: Strengthening Motivation for PA, Also among Inactive Persons

The second theme described how the support should strengthen motivation for PA, especially among inactive persons. As can be seen in Appendix C, included categories related to conscious-raising, making PA enjoyable and being useful for organizations reaching out to inactive seniors.

##### Conscious-Raising

The seniors perceived that making a person aware that his/her current PA in daily life was insufficient could strengthen a person’s motivation for PA. It was envisioned that monitoring and informing the person about current activity and sedentary behavior could contribute to increasing the person’s awareness of the need for behavioral change.
“It might come as a shock, to see the amount of sedentary behavior this time of the year”.


For persons that had started to increase their PA, acknowledgement of progress was described as motivating. Here, both confirmations of different types of PA performed and acknowledgements that personal activity goals had been reached were perceived as valuable. It was also described that some persons were interested in getting confirmations of results on their performance in activities and that confirmations of the health effects from PA (e.g., posture, sleep, and general well-being) were envisioned to promote activity. In addition, the seniors thought that the technology could contribute to strengthening personal trust in physiological signals. For example, continuous pulse-monitoring during exercise could help persons afraid of high pulse to relate a body sensation to an actual pulse. Likewise, objective data from measurements could confirm positive feelings experienced after exercising. This might strengthen peoples’ trust in these feelings.

##### Making Physical Activity Enjoyable

The participants emphasized the solution must make PA more enjoyable, for example by strengthening social interaction. It was described that social interaction could make seniors feel that they belonged to a group and that social interaction could provide social support for PA. Therefore, the participants perceived that the technology should support PA in a group such as group training, playful activities, and competition events containing PA. The seniors gave various examples of existing social activities that included PA which they thought the technology should give users access to. In fact, some participants described that their main motivation for PA was to do PA together with others.
“We used to say that, for example, when we go bowling, that half of the amusement is the bowling and the other half is to get out and socialize with others. That is almost more important. For most of us it is at least more important than getting a good result”.


The seniors also saw that the technology support could use playful elements, competition, and rewards to make PA more fun. For example, games that combine physical and cognitive activity were seen as possibilities for providing the user with different kinds of stimulation, and thereby contributing to variation. Some seniors had arranged a competition where clues found in their outdoor environment had been used in a rebus puzzle. Real-time PA coaching was described as another possible way of making PA more fun. The seniors liked the idea of getting feedback during the activity and saw that the technology could provide some kind of incitement which could stimulate them to work harder. Some persons expressed that they would be interested in positive feedback such as rewards for good performance and for having made new achievements. However, other persons described that they were also motivated by “being whipped at”.
“But then you become motivated to do better in the next training session. I want to get thumbs up again.”


##### Useful for Organizations Reaching Inactive Persons

The participants emphasized that the solution must reach inactive seniors and suggested that different organizations could contribute here. Health care organizations were perceived important for prescribing the solution to seniors in need of increasing their PA. Prescription from a physician, by many seen as an authority, was envisioned to incite patients to use the support, and thereby increase personal PA. In addition, knowing that the measured PA would be discussed in future revisits at the clinic was seen as a motivator for the patients to use the support for increasing their PA. Senior associations and sport clubs were described as additional potential channels for introducing the technology to seniors. It was described that seniors’ motivation for technology use and PA could be strengthened if the technical applications could be combined with the senior group activities organized by the associations. This was seen as an opportunity for the associations to attract new attendees to group activities.
“If we [in the associations] had anything enabling us to attract for example 10–12 persons on our group walks instead of the seven persons that usually come, then we would have made an impact for those who really need the walks.”


#### 3.1.2. HCPs’ Views on Needs and Requirements on Technology-Based Motivation Support for PA

Two themes emerged in the inductive qualitative analysis. Each theme contained three categories, all consisting of 3–10 sub-categories. Themes, categories, and sub-categories are presented in Appendix D and described in detail below.

Theme 1: A Tool for Strengthening the Seniors’ Own Motivation and Ability to Perform PA

The first theme illustrates how the technology should be a tool for strengthening the seniors’ own motivation and ability to perform PA. The HCPs described that the solution should increase the seniors’ own responsibility and control over their PA. Hence, the technology should support self-care performed by the senior independently or in co-operation with a coach. The HCPs emphasized that the technology should motivate healthy seniors for PA and thereby postpone needs for rehabilitation and assistance to the future.

As can be seen in Appendix D, categories in the theme described that the technology should increase the senior’s motivation for PA directly and indirectly by motivating decreased inactivity and also help the senior to overcome hindrances to PA.

##### Increasing Seniors’ Direct Motivation for PA

The HCPs perceived that the technology should contribute to increasing seniors’ insight and knowledge into why and how much they should be physically active. Increased understanding of the aim for PA was envisioned to strengthen motivation for PA and facilitate the HCPs’ clinical work.
“It is still very much about understanding the aim and meaning of PA. Many persons think that a short walk with their dog is enough. But trying to make them realize that they need to increase and be active at another intensity is a challenge.”


Other ways by which the HCPs thought that the technology motivates seniors to perform PA included by stimulating social interaction, by making PA more fun for the senior as well as by monitoring and visualizing the positive effects of performed PA. In addition, the participants thought that the digital support should support seniors’ self-registration of performed exercises. It was also stressed that the technology should support the setting and following-up of functional goals in daily life, for example, being able to go to the bathroom independently. The participants envisioned that increased motivation for PA could prevent passivity among seniors.
“For the seniors we meet at the geriatric clinic, the main benefit might be just to be active during the day. Because an enormous amount of time is spent on nothing, people are very passive.”


##### Helping Seniors Overcome Hinderances to PA

The HCPs perceived that the technology should facilitate seniors in performing PA, for example, by providing visual guidance in exercises and providing reminders to seniors to be physically active and carry out recommended PAs. The participants described fear and doubt about PA as hindrances to PA which the seniors might need support in overcoming.
“Among persons with weak balance that might have fallen. There is often, a not very much expressed, but a certain doubt to be active. And maybe above all else doubts, to be active by oneself.”


##### Increasing Senior’s Motivation for PA through Decreased Inactivity

The HCPs experienced that many seniors were unaware of having an inactive life style and the risks associated with that. It was described that personal attitudes could contribute to inactivity.
“I guess many seniors think that they are worth sitting down, because they have worked and labored all their lives. And this I hear very often from older persons.”


Therefore, the HCPs envisioned that technology-based monitoring of inactivity could clarify risk behavior and push some seniors to be more active. Related to this, the HCPs saw that inactivity monitoring could be relevant for senior patients preparing for planned operations. It was described that rewards for decreased inactivity could be used to promote PA and to strengthen personal motivation for PA. The HCPs emphasized that results from inactivity measurements should be communicated to users in a suitable way.
“It is a form of help to self-help too. Then it might depend a lot on how you approach the person, how it is communicated.”


Theme 2: Useful both for HCPs and Seniors

The second theme illustrates requirements that the technology must meet in order to be useful both for HCPs and seniors. As can be seen in Appendix D, categories in the theme described that HCPs’ view on qualities that makes the technology attractive for seniors, how the technical applications could support their clinical work, and how the technology should facilitate co-operation between senior and professional.

##### Attractive for Seniors

The HCPs perceived that seniors who experience improved well-being when using the technology will be motivated to continue use. Likewise, confirmations of progress made during periods of technology use was perceived to motivate technology use. It was emphasized that the technology should be fun to use and make PA fun. Some HCPs described solutions they had seen enabling patients to bike in a virtual environment, they thought this type of stimulation during PA could motivate users. Attractive design was also envisioned to contribute to users’ feeling modern and to increase motivation for technology use. The HCPs also stressed that, in order to be useful, the technical applications must be easy to use and understand, as well as work stably and be handy to carry around. It was also stated that the applications must be adaptable. Moreover, the participants described that the technology must support safe and secure data management that enables the user to retrieve data in a smooth way.
“My experience is that seniors used to say that they want to feel that they own the data themselves. That they want to have control of the information.”


It was expressed that technology should be interoperable with other systems in order to be attractive for seniors. In relation to this, it was seen desirable if the technology was able to work stably without additional equipment or wireless internet access at home. Finally, the HCPs saw that distribution channels and business models could affect the seniors’ motivation for using technology. For example, some people might be more motivated to use a consumer product that they had bought themselves. However, the participants suggested that the applications should be free of charge or very cheap for senior users.

##### Supporting the HCPs’ Clinical Work

The HCPs expressed that, in order to support their clinical work, technical applications must have a clear aim and a proven long-term effect on the senior patients’ motivation for PA. They also emphasized that the technical applications must be credible and inspire confidence: as professionals, they wanted to feel that they could really support and recommend the applications to seniors.

The HCPs discussed that it is individual whether PA monitoring can support PA. They stressed that the technical applications should be customizable according to the individual user’s needs, preferences, and situation. It was emphasized that the applications must be customizable to the aim of the individual’s PA and support the specific type of PA that the clinician and patient are focusing on. Likewise, the HPCs stated that the technology should support setting individual and flexible goals.
“If we talk about a target group that needs rehabilitation in order to be activated, then it can be a real failure if you have a really bad day and are not able to do anything. In that situation, to get digital feedback saying today you have accomplished nothing. When instead I could already, from the start, lower my goal. Because then I might feel cheered up despite the bad day with low energy. Because I did something. And I think that is important.”


The HCPs also stated that, in order to be useful in their clinical work, the technical applications and devices must fulfil requirements from the health care organizations, for example, requirements on hygiene and economy. Finally, the professionals discussed whether technology could be useful also for persons with cognitive failure. However, they perceived that in order to be useful for this group, digital technical applications need to be further adapted and also used with extra support from a person.
“I think that there is also a downside to this, a risk I can see is if the technology is used in order to replace any form of human contact. Especially when it comes to the oldest seniors with cognitive failure, then it is important to get another form of support, otherwise the technology will only become an obstacle.”


##### Facilitating Dialogue and Cooperation between Senior and Professional

The HCPs saw that the technology could be very useful in their clinical work if it supported their dialogue and cooperation with the seniors.
“Seen to the work process of a physiotherapist, if you work with a patient for a longer period of time, it is always good to see if there has been an actual change. And in that sense, it might be difficult for some individuals to actually describe themselves if they have done something significantly different or not.”


In particular, they stressed that the technology should strengthen the seniors’ active participation in the process, something they considered highly important for the result. They also saw the technology should provide objective measurement data and support following-up at a distance.
“It might facilitate, I also think, if you are going to evaluate something. If you for example give the person an exercise program or a process and then the person says, ‘I have done a lot of training every day’, and then you see no progress. ‘Well, you haven’t been physically active according to this’. Then you can sort of discuss—is the lack of progress due to the fact that the exercises given are irrelevant or due to the low activity. So, it might facilitate our evaluation.”


### 3.2. Analysis of Similarities and Differences between Seniors’ and HCPs’ Views

Two dimensions evolved in the analysis of sub-categories from the qualitative analyses of focus group interviews with seniors and HCPs, respectively: the former reflected seniors’ and HCPs’ views on how the technology should contribute to increasing seniors’ PA and the latter illustrated qualities that the technical support should have in order to suit seniors and coaching professionals.

#### 3.2.1. Views on Possible Contributions from the Digital Technology in Supporting and Motivating Seniors to Increase PA

A detailed presentation of the seniors’ and HCPs’ views (sub-categories) in this dimension is presented in Appendix E. As can be seen in Appendix E, both seniors and HCPs described that the technical support should make PA more enjoyable. Here, social interaction was mentioned by HCPs and strongly emphasized by seniors. The seniors also perceived social games as one possible way of making PA more fun. Moreover, both seniors and HCPs perceived that the technical applications should provide support to the senior. However, the seniors described that support could come either directly from the applications, from associations or health care, while the HCPs focused mainly on how the applications could support their work with coaching seniors in PA. The HCPs described different ways by which they saw that the technology could support this. The seniors also stated that support for technology use must be available.

The HCPs emphasized that the technology should strengthen the seniors’ control of the process of PA behavioral change to increase PA. Therefore, they saw that technical applications should contribute to increase seniors’ knowledge of PA as well as to support self-monitoring. The seniors, on the other hand, primarily described that the technology should provide feedback on activity level and progress, whereas the HCPs envisioned that the technology could support goal setting and help to clarify risk behavior.

Also, the HCPs described that, in order to be appealing for them as professional users, the technology must contribute to the well-being of seniors by making the seniors trust the solution and by contributing to the seniors feeling modern and confident. The HCPs’ views on which factors are important for seniors’ well-being might not be generalizable to all HCPs and seniors. However, they reflect experiences and perceptions of the HCPs participating in the focus group interview. The HCPs described that they, also, in their professional role needed to believe in the solution so that they could truly recommend it. These descriptions were not found in the sub-categories of the seniors’ views. However, both seniors and HCPs expressed that the technology must be perceived as supportive and non-demanding in order to make the seniors feel better when using the application. Moreover, both seniors and HCPs perceived that the technology needed to contribute to feelings of safety in the sense that the technology itself must not provoke fear and also should contribute to decreasing seniors’ fear and doubt about PA. Finally, the HCPs envisioned that the solutions should be enjoyable and provoke emotions of pleasure among seniors. The view that the technology itself should be fun was not identified among the sub-categories of the seniors’ perceptions.

#### 3.2.2. Views on Qualities that the Digital Technology-Based Motivational Support Should Have

A detailed presentation of the seniors’ and HCPs’ views (sub-categories) in this dimension is presented in Appendix F. As can be seen in Appendix F, seniors and HCPs expressed similar views on the qualities that the technical motivation support must have in order to be attractive and usable for users: easy to use and understand, reliable, customizable, interoperable, and effective for the purpose were expressed by both user groups. In addition, the HCPs described that the technical support must be low-cost for the user and also accessible in a way that motivates the user to use it. For example, some persons might be more motivated to use a consumer product that they have bought themselves.

## 4. Discussion

This study illustrates the views of seniors and HCPs on digital technology-based motivation support for increasing seniors’ PA. Results from the study showed that both seniors and HCPs see several possible contributions from technical support and have similar views on which qualities the technology must have to be attractive and useful. Both seniors and HCPs saw that technology should support PA and make it more enjoyable while strengthening the seniors’ control and well-being. However, the seniors´ opinions were related to social aspects, enjoyment, and how the technology could contribute to making them feel better. The HCPs views also included aspects on how the technology could contribute to their role as professional coaches and how the technology could facilitate the dialogue between the senior and coaching HCP as well as providing information. Both seniors’ and HCPs’ views can be supported by the SDT. Also, BCTs relevant for increasing seniors’ PA can be identified in the results. Four main contributions of the technology were identified in supporting and motivating seniors to increase PA; they are discussed in the following section both from a theoretical perspective and in relation to previous research in the area.

Firstly, seniors and HCPs perceived that the digital technology should make PA more enjoyable. This can be seen as a way of raising the seniors’ intrinsic interest for PA, which is in accordance with SDT, and especially, its sub-theory cognitive evaluation theory (CET) [26]. Examples provided on how to raise intrinsic interest were to include playful and competitive elements in the technical applications and to make the technology support social interactions. Especially, the seniors emphasized social interaction as a main motivator for PA. Social support has also been found to improve the effectiveness of PA interventions [34]. Concretely, social interaction, such as group activities that include PA, could contribute to satisfying the psychological need of relatedness which, according to SDT, can contribute to intrinsic motivation and to autonomous motivation for PA. The importance of social interaction for behavioral change is also in accordance with the social cognitive theory which emphasizes the importance of social interaction and model learning [35]. However, this theory also describes that personal, behavioral, and environmental factors interact with each other in behavioral change. All three factors are, therefore, necessary for behavioral change [35]. In addition, the technical support could contribute to reducing loneliness among seniors. It is also postulated in SDT that satisfaction of the psychological needs of competence and autonomy is important for a person’s autonomous motivation. It should be emphasized that the senior participants in the focus group were physically active, some of them were also used to PA in a social setting. For persons that do not feel competent and autonomous in a PA group setting, technical applications can contribute to satisfying their need for relatedness, for example, by providing different social features. Social interaction could also be relevant for introducing new users to the technology. For example, UTAUT describes that a system is perceived more attractive to new users when used by other persons they have confidence in, and that access to help from other users with the usage of the system can strengthen personal motivation for using the system.

Secondly, both seniors and HCPs described that the digital technical applications should provide support for PA. Here, the HCPs described how technology could facilitate their role as professional coaches. However, the seniors expressed a broader view on coaching and saw that coaching could be provided from the technology itself, from senior organizations, as well as from health care. The seniors thought that some inactive persons could become motivated to increase their PA after prescription by a physician and follow-up meetings. Some of the HCPs gave similar descriptions of their work procedures together with patients. Provision by a professional can, according to SDT (particularly its sub-theory organismic integration theory (OIT), which describes the process through which a person can become more intrinsically motivated to a behavior that he/she initially was amotivated) be seen as the least autonomous form of extrinsically motivated behaviors performed to satisfy an external demand or obtain external reward or avoid external punishment. According to SDT, contextual factors can support internalization and integration of regulations, which are important processes for intrinsic motivation, which in turn, contributes to long-term behavior maintenance. Also, coaching can contribute to satisfying the senior’s need of relatedness while gradually increasing satisfaction of personal needs for competence and autonomy. According to SDT, this process contributes to autonomous motivation and sustained behavior.

Thirdly, it was expressed that the digital technology should contribute to strengthening the seniors’ own control in changing PA behavior. This view is supported by SDT which states that satisfying the personal need for autonomy is important for strengthening autonomous motivation and sustained behavior. The HCPs perceived that seniors’ motivation for PA could be strengthened if the technology informed and raised the seniors’ awareness of the importance of PA. According to OIT, being physically active in order to avoid health risks is the least autonomous form of extrinsically motivated behaviors. Organismic integration theory states that, in order to strengthen the individual’s autonomous motivation for PA, promotion of internalization of values and behavior related to being physically active is needed. This was illustrated by the HCPs statement that results need to be communicated by the applications in a suitable way. Moreover, the refined taxonomy of BCT to help people change their PA includes two BCTs on providing information on consequences of behavior (in general or behavior to the individual) [23]. However, providing information of consequences on behavior has not been identified as among the most efficient BCTs for increasing seniors’ PA [36]. The HCPs also perceived that the technology should provide visual guidance. This is interesting since two previous studies have pointed out that, although being effective for increasing PA, BCTs related to practice, planning, problem solving, and providing instructions are scarce in current activity monitoring applications [37,38]. Also, the HCPs described different ways by which they wanted the technology to support seniors’ self-monitoring. Previous studies have shown that BCTs related to self-monitoring and self-regulation are most frequently found in activity monitor systems [37,38]. However, BCTs related to self-regulation have not been identified among the most efficient BCTs for increasing seniors’ PA [36]. On the contrary, there is evidence that later in life, people tend to prioritize present-oriented goals related to emotional meaning over future-oriented goals [39]. Therefore, it might be important in information aiming at increasing seniors’ motivation for PA, to focus on the short-term benefits of PA and to include suggestions for emotionally meaningful activities to seniors. The seniors perceived that motivation for PA could be strengthened by personal awareness of insufficient PA in their daily life and acknowledgement of progress. The latter is interesting since rewards for successful behavior have been identified as one of the three most effective BCTs for increasing older adults’ PA [40]. Moreover, the seniors perceived that the technical support should motivate the user to perform PA by providing feedback. The feedback can be seen as support for the users’ internalization and integration of regulations, which, according to SDT, can increase the degree of self-determination in PA behavior and thereby enhance persistence and adherence.

Fourthly, it was expressed that the digital technology should contribute to strengthening well-being. The seniors described that the technology must be perceived supportive, non-demanding, and must not provoke fear of making mistakes (for ruining something, and for needing help). The described aspects relate to key constructs of UTAUT (expectancy of the system’s performance, effort needed for using the system, and degree of ease to use a system) which are important for technology acceptance and use according to UTAUT [20]. The seniors’ view that the technology should be perceived safe and supportive is also in accordance with SDT, which states that fulfilment of psychological needs for competence and autonomy is relevant for increasing the seniors’ autonomous motivation for technology use and for PA. Moreover, the HCPs perceived that the technology should contribute to decrease seniors’ fear and doubt for PA. This view is both in accordance with SDT, which describes that satisfying the psychological need of competence can contribute to strengthening autonomous motivation, and with a study by French et al. [40] which identified BCTs related to problem solving as one of most effective BCTs for increasing older adults’ PA.

Both HCPs and seniors expressed that the digital technology should make seniors feel better. The HCPs also perceived that the technology should provoke feelings that could increase the seniors’ intrinsic interest for the technical applications, for example, feelings of enjoyment and being modern. According to SDT, this is relevant for strengthening seniors’ intrinsic motivation to use the devices. Finally, the HCPs described that users must feel confident in the technology. For example, they described that technical applications must communicate results to seniors in a suitable way, which is in accordance with SDT, describing that satisfying the needs of relatedness is important for autonomous motivation. The HCPs also emphasized that, as professionals, they need to feel they can truly recommend the technical support to patients. This view is supported by UTAUT describing that users’ expectancy on system performance is important for technology acceptance and use.

Seniors and HCPs had similar views on what qualities the digital technology must have to be attractive and useful: both user groups expressed that the technical applications should be user friendly, reliable, customizable, interoperable, and effective for the purpose. All mentioned aspects are relevant for satisfying the users’ needs of autonomy and competence, and therefore in accordance with SDT which states that this is important for autonomous motivation to use the devices [26]. Moreover, the mentioned qualities are also supported by UTAUT as determinants for technology acceptance and use [20]. The HCPs also described that price and distribution channels could be relevant for the seniors’ motivation to use the support. This view is in accordance with UTAUT, which describes facilitating conditions as having a positive effect on users’ willingness to accept and use technology.

To summarize, the identified requested contributions of the digital technology are relevant for strengthening motivation according to SDT [41], since they reflect needs and requirements on a system that can be intrinsically interesting for users and that can increase the users’ motivation quality for PA. Moreover, it illustrates how the system can contribute to the satisfaction of seniors’ psychological needs for relatedness, autonomy, and competence, which according to SDT, can increase intrinsic motivation for PA and thereby strengthen adherence and persistence. The results also contain examples of BCTs that are relevant for increasing PA behavior as well as needs and requirements important for technology acceptance and use according to UTAUT [20].

The results from this study are in accordance with empirical studies on requirements on technology supporting PA. For example, the contributions and qualities identified in this study overlap with the design requirements (giving users proper credits for activities; providing personal awareness of activity levels; supporting social influence; and considering the practical constraints of users’ life styles) identified by Consolvo et al. [42]; core features (identified up-to-date and evidence-based information and instructions, self-regulation tools, social interaction, personalized set-up, attractive design and content, and access to the internet service) according to people with rheumatoid arthritis identified by Revenäs [43]; requirements expressed by HCPs and person’s with cardiovascular disease (patient tailoring, simplicity within the platform, technology-augmented care, enabling or increasing individual self-management, and capitalizing on an appropriate time to intervene in the rehabilitation) identified by Walsh et al. [44]; and with design considerations (functional exercises, behavior change and blended technology delivering programs through interactive video) and requirements (comprehensiveness, effectiveness, adaptability, and remote guidance) for an intervention supporting seniors’ exercise at home with help of technology identified from scientific literature and consultancy from experts from health and behavior science, respectively, in a study by Mehra et al. [45].

Results from this study can be used for further exploration of seniors’ and HCPs’ views on hindrances and motivators for PA and for designing interventions for increasing seniors’ PA as well as for technical systems supporting and delivering interventions. In addition, this study also aimed to relate seniors’ and HCPs’ views on theory for personal motivation for PA (SDT) and technology acceptance and use (UTAUT). Thereby, the study might contribute to increased understanding on concrete ways to build technology-based motivation support for seniors’ PA based on theory for personal motivation. Moreover, areas where seniors and HCPs expressed different views on approaches for increasing seniors’ motivation for PA were identified. Further research is needed on potential differences between HCPs’ and seniors’ views on how to strengthen seniors’ motivation for PA. In particular, the third contribution identified in this study (i.e., strengthening the seniors’ own control) is relevant for further exploration, especially since the HCPs’ view (that increasing seniors’ awareness of risks associated with low PA and how active they need to be) according to SDT will contribute to the least autonomous form of extrinsically motivated behaviors. Here, it might be relevant to relate the results to complementary theories for PA and health behavior. For example, theories including the key construct of self-efficacy [46] such as the social cognitive theory [35] and the health belief model [47]. Exploration of individual barriers and facilitators for participation in fall prevention programs have been suggested to improve HCPs’ possibilities to build partnerships and develop person-centered care [48]. Further research is also needed on how technical support can contribute to seniors’ internalizing and regulating of integration. For example, how can technology facilitate the dialogue between HCP and senior, as requested by the HCPs in this study. Further investigations on how new types of BCTs can be integrated into PA support systems is also needed. For example, BCTs for relapse prevention and prompt barrier identification are most often not included in mobile applications [49].

The strengths of this study include the representation of two different potential user groups (seniors and HCPs) in early phases of a user-centered design process. Investigating both user groups’ needs prior to technology development is important since it enables the design and development of a system that is usable for both groups. Furthermore, potential conflicts between the two user groups’ needs can be identified. Moreover, HCPs represented different organizations, different professions, and professional roles. In addition, to ensure the trustworthiness of the result, the analysis was performed by two researchers and read by a third researcher with previous experience in qualitative research and seniors’ PA. Accordingly, the papers discussion section has been reviewed by an external researcher with deep knowledge in SDT. The paper provides a detailed description of how data was collected and analyzed. Moreover, the transparency of the analysis has been strengthened by citations from the interviews and description the participants groups and research team.

The main limitations of the study included the rather limited sample due to the fact that only one focus group was conducted with seniors and one with HCPs. Another limitation was the selection bias since the senior participants were active persons in independent living. Therefore, the results of the study might not be transferable to inactive seniors, persons with transportation barriers, persons living in special housing or in rural communities. The technology might be helpful also for those persons and further investigation is needed in this area. However, future work will also involve inactive seniors in the development process, where needs and requirements for technology supporting seniors’ PA will be further explored. Also, some of our senior participants had experiences from leading and arranging social activities and training activities for seniors. Some of their views described needs and requirements that they perceived important for other seniors that they had met in those roles. Moreover, the analysis of similarities and differences among the two user groups was performed on sub-categories from the two analyzed focus group interviews. Hence, the data subjected to the second analysis were sub-categories from the first step of qualitative analysis containing interpretation by the researchers. Therefore, the researchers’ perceptions might have influenced the data used for the second analysis and this might affect the results. However, interpretation is always a part in qualitative analysis, for example, as described in Silverman [50].

## 5. Conclusions

Seniors in the study perceived that digital technology-based motivation support for PA needs to be both a help in daily life and a motivator for PA. The importance of social interaction for increasing motivation for PA was emphasized.

The HCPs in the study perceived that digital technology can support and motivate seniors to increase PA, for example by being a tool that increases seniors’ motivation and ability to perform PA and by facilitating co-operation and dialogue between senior and HCP. The HCPs also stressed that the technology-based support needs to be useful both for seniors and HCPs. In addition, the HCPs described that the technical application should increase the seniors’ consciousness of PA and provide objective information on PA behavior.

Both seniors and HCPs perceived that the digital technology should make PA more enjoyable, provide support, strengthen the seniors control, and contribute to well-being. However, seniors and HCPs expressed different perspectives on how the technology concretely could contribute. While seniors emphasized enjoyment and social interaction, HCPs also highlighted support in their professional role as coaches of PA. Both the seniors’ and HCPs’ views are in accordance with SDT.

The seniors and HCPs had similar views on qualities that the digital technology should have in order to be useful and attractive. The expressed views are supported by UTAUT.

## Figures and Tables

**Figure 1 ijerph-16-02418-f001:**
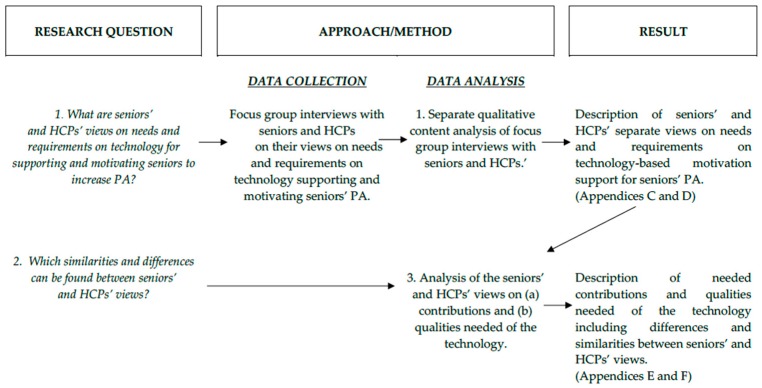
Overview of the study design.

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
