# Peer review of "Technology-Based Motivation Support for Seniors’ Physical Activity—A Qualitative Study on Seniors’ and Health Care Professionals’ Views"

_ijerph, 2019, doi:10.3390/ijerph16132418_

Round 1

Reviewer 1 Report

The study, “Technology-based motivation support for seniors physical activity-a qualitative study on seniors and health care providers views” presents results from a sample of seniors and caregivers. While some parts of this manuscript were interesting, other areas could be improved. I hope the authors consider my feedback for enhancing their paper.

·         Participants and Recruitment: More details would be appreciated here. How specifically were participants recruited (word of mouth, asking persons directly, etc.)? A breakdown of any that refused participation should also be mentioned.

·         Discussion and Conclusions: Considering how the sample was recruited, n=, and demographic factors, how the results are referred to should be tapered a bit. For example, in lines 576-578, these results are not generalizable to all seniors and HCPs. Revise where appropriate.  

·         If any information is available, it might be nice to include role of feasibility for seniors, HCP, and investigators (PubMed ID: 19362699).

·         While these qualitative results could have ranging interpretations, this reviewer would like to see some discussion points about the role of technology on reducing loneliness. How might this platform bring relatedness to seniors? This is an important part of exercise psychology (as was mentioned somewhat already).

·         Technology might also be helpful for seniors that have transportation barriers and live in rural communities?

·         Selection bias should be highlighted as a limitation.

·         Line 37: Should be, “at least three hours of exercise per week”.

·         Line 46: Should be, “Increased”.

·         Line 49: Should be, “concluded”. Be sure to correct for minor grammatical mistakes throughout.

·         Make any changes to the abstract that align with those made in the text.

Author Response

RESPONSE TO COMMENTS FROM REFEREE 1

General comments:The study, “Technology-based motivation support for seniors’ physical activity-a qualitative study on seniors and health care providers views” presents results from a sample of seniors and caregivers. While some parts of this manuscript were interesting, other areas could be improved. I hope the authors consider my feedback for enhancing their paper.

Response:We are grateful to your constructive feedback which has been very helpful in our work to enhance the paper. 

Comment 1:Participants and Recruitment: More details would be appreciated here. How specifically were participants recruited (word of mouth, asking persons directly, etc.)? A breakdown of any that refused participation should also be mentioned. 

Response: We have now included information about the recruitment process of seniors; please see green text paragraph 2.3.1, lines 155-164. Likewise, we have completed with information about the recruitment process of HCP´s; please see green text in paragraph 2.3.2, lines 188-195.

Comment 2:Discussion and Conclusions: Considering how the sample was recruited, n=, and demographic factors, how the results are referred to should be tapered a bit. For example, in lines 576-578, these results are not generalizable to all seniors and HCPs. Revise where appropriate.  

Response: We agree that the referred to are not generalizable to all seniors, for example the HCPs’ view on how the technology should contribute to the well-being of seniors. To clarify this, we have added an extra sentence after this view (which is referred to in the comment, previously found on lines 576-578). Please see green text on lines 514-516: “The HCPs’ views on which factors are important for seniors’ well-being might not be generalizable to all HCPs and seniors. However, they reflect experiences and perceptions of the HCPs participating in the focus group interview. “

Also, this is further elaborated in the methods discussion section, please see line 710-712.

Comment 3: If any information is available, it might be nice to include role of feasibility for seniors, HCP, and investigators (PubMed ID: 19362699). 

Response:

We had difficulties in understanding this comment. Our interpretation is that the referee asks for a description of the roles of seniors, HCPs and investigators for the feasibility of the study. Therefore, we have included a brief description of this in the introduction section (please see green text on lines 88-90) and in section 2.1 (please see green text on lines 126-127).

If you see that further clarifications or modifications are needed, we are willing to contribute further.

Comment 4:  While these qualitative results could have ranging interpretations, this reviewer would like to see some discussion points about the role of technology on reducing loneliness. How might this platform bring relatedness to seniors? This is an important part of exercise psychology (as was mentioned somewhat already).

Response: We agree that it is relevant to discuss the role of technology on reducing loneliness in relation to the psychological need for relatedness. This has been expressed in the current version of the text (please see green text on lines 555-557): “By supporting social interaction and contributing to satisfaction of the psychological need of relatedness, the technical support could contribute to reduce loneliness among seniors.” 

Comment 5:  Technology might also be helpful for seniors that have transportation barriers and live in rural communities?

Comment 6:  Selection bias should be highlighted as a limitation.

Response to comment 5 and 6: We agree that the technology also might be helpful for seniors that have transportation barriers and live in rural communities. These aspects are now explicitly mentioned in the discussion section describing main limitations. Moreover, we have highlighted selection bias as a limitation (please see green text on lines 708, 710-712). Current version is: “Another limitation is the selection bias since the senior participants were active persons in independent living. Therefore, the results of the study might not be representative and transferable to inactive seniors, persons with transportation barriers, persons living in special housing or in rural communities. The technology might be helpful also for those persons and further investigation is needed in this area.

Comment 7:  

Line 37: Should be, “at least three hours of exercise per week”.

Line 46: Should be, “Increased”.

Line 49: Should be, “concluded”. Be sure to correct for minor grammatical mistakes throughout.

Response to comment 7:The three grammatical mistakes have been corrected accordingly and the whole manuscript has been proof read by research colleague skilled in written English.

Comment 8:  Make any changes to the abstract that align with those made in the text

Response:We have added the word “digital” to technology support to further clarify what we mean with technology, as pointed out by referee 2. Smaller modifications of the text have been performed, they are all colored green.

Reviewer 2 Report

Technology-based Motivation Support for Seniors’ Physical Activity – A Qualitative Study on Seniors’ and Health Care Professionals’ Views. The main aim was to investigate seniors’ and health care 79 professionals’ (HCPs’) perceptions on possible contributions and qualities needed/required from technology in supporting and motivating seniors to PA. A secondary aim was to discuss whether the views expressed were in accordance with the SDT, UTAUT  or reflected elements BCT taxonomy for PA behavioral change. 

The topic is of great importance. The aims are well described and answered in a satisfactory way. I have some minor suggetions/comments. I miss a short description of Your definition of technology, as aslo pencil and paper is a form of (old) technology. Perhaps You mean digital technology? The SDT is used as a theoretical background, with good result, but I miss other theories that might be useful, especially theories of health behaviour. The arguments in line 623-625 could have been expanded with the Knowledge Practice Attitude (KAP) model, as an example. Also the role of important other and self-efficay could have been used in several of the discussing points. Also, much research from David Markland, on physical activity and SDT could have been used. See for example https://ijbnpa.biomedcentral.com/track/pdf/10.1186/1479-5868-9-78.

I do not see lack of representation as a serious weeknes in this study, in qualitative research it is better with a strategic sample, which You have, than a representative.

Parts of what is written in 3.1.1 and 3.1.2 (line 197 to 223) may be moved to the Methods part, as they are not results in my view. Text in  line 456-463 also is repeated in line 503-508, especially the citation.Check wording in line 577 (...from technical...).

Author Response

RESPONSE TO COMMENTS FROM REFEREE 2

General comments:Technology-based Motivation Support for Seniors’ Physical Activity – A Qualitative Study on Seniors’ and Health Care Professionals’ Views. The main aim was to investigate seniors’ and health care 79 professionals’ (HCPs’) perceptions on possible contributions and qualities needed/required from technology in supporting and motivating seniors to PA. A secondary aim was to discuss whether the views expressed were in accordance with the SDT, UTAUT or reflected elements BCT taxonomy for PA behavioral change. 

The topic is of great importance. The aims are well described and answered in a satisfactory way. I have some minor suggetions/comments. 

Response to general comments:We are pleased to hear that you find the topic important and have found aims described and answered in a satisfactory way. We are also grateful for your comments and suggestions which has been very helpful in our work to enhance the paper. 

Comment 1:  

I miss a short description of Your definition of technology, as aslo pencil and paper is a form of (old) technology. Perhaps You mean digital technology?

Response:That is a good point. In order to clarify our definition, we have added the term “digital” to the text on several positions in the manuscript (colored green). The word digital has also been included in figure 1 and in appendices C-F.

Comment 2:

The SDT is used as a theoretical background, with good result, but I miss other theories that might be useful, especially theories of health behaviour. The arguments in line 623-625 could have been expanded with the Knowledge Practice Attitude (KAP) model, as an example. Also the role of important other and self-efficay could have been used in several of the discussing points. Also, much research from David Markland, on physical activity and SDT could have been used. See for example https://ijbnpa.biomedcentral.com/track/pdf/10.1186/1479-5868-9-78. 

Response:

We agree that it might be useful to relate the study results to other theories, especially for PA behavior. Therefore, we have clarified the presentation of an article by Rhodes et al. describing theories for PA behavior in the introduction (please see green text on lines 70-71). Moreover, we have clarified in the discussion that future studies (especially on differences between HCPs’ and seniors’ views on how to strengthen seniors’ motivation for PA which are relevant for the arguments exemplified in the comment) should relate to complementary theories for PA and health behavior. Here, we have especially mentioned theories including key construct of self-efficacy and given social cognitive theory and health belief theory as examples of useful theories (please, see green text on lines 684-687). Moreover, we have related some of the results to the social cognitive theory.

Our main motif for choosing SDT as main theory to relate to in the paper is that SDT’s strong support in PA behavioral change research literature, especially for long-term maintenance. We are grateful for the suggested work by David Markland and have included the suggested article by Teixeira et. al as a reference (number 28) which is referred to in the introduction (please, see green text on lines 79-80). We find also the Knowledge-attitude-behavior model interesting but have decided not to relate our results to this model, mainly because of criticism and weak support in research literature on behavioral change in related areas (for example https://onlinelibrary.wiley.com/doi/epdf/10.1038/oby.2003.222).

Comment 3:

I do not see lack of representation as a serious weeknes in this study, in qualitative research it is better with a strategic sample, which You have, than a representative.

Response: We share the view that it is not a serious weakness. However, we still think it is worth mentioning lack of representation as a weakness study and have therefore kept it there (this was suggested by another referee).

Comment 4:

Parts of what is written in 3.1.1 and 3.1.2 (line 197 to 223) may be moved to the Methods part, as they are not results in my view. Text in line 456-463 also is repeated in line 503-508, especially the citation. Check wording in line 577 (...from technical...).

Response to comment 4: We have moved text in sections 3.1.1-2 to the Methods part (please see green text on lines 166-181 and on lines 196-204). Moreover, we have removed the repeated text (about views technology for persons with cognitive failure) which in the earlier version of the manuscript was found twice. This text is now only found at lines 468-473 (please, see green text there).

Finally, the word “support” has been added to the wording (...from technical...) which was pointed out (previously on line 577).

Reviewer 3 Report

Introduction

92-100 This part of the introduction is already giving away a short summary of results. I would rather just end this chapter with the two research questions.

2. Materials and Methods

2.1 Study design

The study design reveals methodological weaknesses. One the one hand you are writing about a descriptive qualitative content analysis, on the other hand you are talking about analyzing the content of the focus group inductively. First, a pure descriptive way of analyzing data contradicts an inductive approach, where emerging topics in the data material are detected and clustered into codes, sub-codes etc. Second, interpretation is a very important part of qualitative research and cannot simple be eroded by a descriptive way of analyzing the data material. A further source might be helpful here, i. g. “Interpretation in Qualitative Research”: https://us.sagepub.com/sites/default/files/upm-binaries/82114_Willig_Rogers_Ch16_2p.pdf

Thirdly, it is not clear what kind of methodical “tool set” you were concretely using. Qualitative content analysis is widely used in social sciences, but different authors use very different tool sets. Margrit Schreir (2012) published a book on “a comparative content analysis in practice”, which might be usefull for you, especially in the context of your 2nd research question.

Furthermore, the manuscript misses clarification on how you (technically) analyzed your data (which program or method did you use?  Please also name the software or reference a “paper and pencil”-method (there is literature on that as well).

116-120: Stylistically, this section would probably better serve as an introduction to chapter 2.

2.2 Ethical considerations

129-132 This statement seems a bit casual: since research is conducted inductively some personal or emotional topics might emerge in focus groups. Focus groups should be understood as a very sensitive way of collecting qualitative data, where it is even harder to protect people’s anonymity, since other participants might talk about the group and about other participants.

2.4 Data Collection

170: It would be nice to know more on background and experience on focus group moderator.

176: Rather use the term “group interviews” than “interviews”.

2.5.1 Description of the Participants

In my opinion, this sentence/statement could be merged into chapter 2.5.2

Results

3.2. This chapter should be patched together with the methods chapter, it seems misplaced and redundant here.

Helpful facilitator

300-307 Some more background information on this code “would be useful. How precisely did they express that they needed to feel technology contributing to their well-being? Do they also “blame” themselves for needing technology to give them a push?

Conscious-raising

332-334: Did participants elaborate on these “positive feelings”?

3.2.2. HCPs’ views on needs and requirements on technology-based motivation support for Past

Instead of themes, categories and sub-categories I would suggest a different hierarchy: Categories, sub-categories, dimensions. It was helpful to see that you included a table where coding system is presented.

Generally, results of the qualitative data need to be presented in a less descriptive way. Topics/categories/sub-categories should first be presented (as in table) in a self explanatory way. Participant’s quotes can definitely be added for a fuller understanding, but should be used more carefully all over the results part. Moreover, it is not clear if these quotes reflect just a single opinion of a participant or if they are representing all participants as a form of an anchor quote. 

415-423, I would suggest to better connect different aspects of the categories/sub-categories. How exactly do arguments differ from one another? Paragraph should be re-written stylistically as well, i.g. you often use terms such as “moreover”, “in addition”.

506: There was only one part of the quote italic, what does this mean?

3.3. Analysis of similarities and differences between seniors’ and HCPs views

550-553: This section seems to belong into the method’s chapter.

Increase the senior’s direct motivation for PA

404-407: This quotation could be shortened in order to reach a “thicker description”.

Increase the senior’s indirect motivation for PA through decreased inactivity

434-451: This whole chapter could be merged with the chapter on “increase the senior’s direct motivation for PA”. A “thicker description” in the way of Clifford Geertz would help here to present both a direct and indirect motivation for PA.

Support the HCPs’ clinical work

Generally, please shorten this chapter and present the narratives more to the point.

522-530: Where does this quote end?

Discussion

610-626: It is very important that you bring theory into the discussion and connect it to your data. Technology should make PA more enjoyable, therefore senior’s intrinsic interest for PA should be investigated and are important, but are senior’s intrinsic motives the only way to make PA more enjoyable? Later you are also stating that “social interaction” is important for seniors, but there’s missing substantial social theory to this point. It is necessary that you include a theory on social interaction and connect it to categories in your data.

728-774: Maybe an under chapter on the strengths and limitations of the study would help to structure the text?

Conclusion

776: Conclusion could start with 777.

Author Response

RESPONSE TO COMMENTS FROM REFEREE 3

General response to the referee:We are grateful to your constructive feedback which has been very helpful in our work to enhance the paper. Your feedback on methods and results of the qualitative analysis has further developed the manuscript.

Comment 1:

Introduction

92-100 This part of the introduction is already giving away a short summary of results. I would rather just end this chapter with the two research questions

Response:

We have followed the instructions to authors in the IJERPH Microsoft Word template which says “Finally, briefly mention the main aim of the work and highlight the principal conclusions”. Therefore, we have kept the sentences in the introduction which highlights the principal conclusions.

Comment 2:

2. Materials and Methods

2.1 Study design

(a) The study design reveals methodological weaknesses. One the one hand you are writing about a descriptive qualitative content analysis, on the other hand you are talking about analyzing the content of the focus group inductively.

First, a pure descriptive way of analyzing data contradicts an inductive approach, where emerging topics in the data material are detected and clustered into codes, sub-codes etc.

Second, interpretation is a very important part of qualitative research and cannot simple be eroded by a descriptive way of analyzing the data material. A further source might be helpful here, i. g. “Interpretation in Qualitative Research”: https://us.sagepub.com/sites/default/files/upm-binaries/82114_Willig_Rogers_Ch16_2p.pdf

Response:

We agree and have decided to describe our method in a more stringent way to make it clearer for the reader what kind of methods we have used. Therefore, we have made the following modifications in the text: The term “descriptive” has been exchanged with “explorative” (please see line 114). In order to further increase the stringency in the description of the method we have added the word “content” on line 120. We have followed  Graneheim and Lundman’s description of content analysis): While the codes were held close to the interview texts (as described by Graneheim), the categorization contained interpretation and was done in jointly by two researchers. We hope that the text modifications have clarified this. We are grateful for the literature suggestion.

(b) Thirdly, it is not clear what kind of methodical “tool set” you were concretely using. Qualitative content analysis is widely used in social sciences, but different authors use very different tool sets. Margrit Schreir (2012) published a book on “a comparative content analysis in practice”, which might be usefull for you, especially in the context of your 2nd research question.

Response:

This comment is highly appreciated by the authors who have had extensive discussion about the analysis related to research question 2. Unfortunately, we were not able to get access to the suggested book by Margrit Schreir within this short time frame. However, we have consulted other articles on methodology (including Fielding and Schreir) and on studies applying comparative content analysis. 

Our conclusion is that our analysis method is purely qualitative and therefore we have revised sections 2.5.1 and 2.5.2 in order to be more explicit and stringent in our description of methodology used according to Graneheim.

(c) Furthermore, the manuscript misses clarification on how you (technically) analyzed your data (which program or method did you use?  Please also name the software or reference a “paper and pencil”-method (there is literature on that as well).

Response:

We have clarified how we technically analyzed the data. The description is included, please see green text on lines 231-238 and line 256.

(d) 116-120: Stylistically, this section would probably better serve as an introduction to chapter 2.

Response:

We agree, this section is now an introduction to chapter 2, please see green text on lines

 and have moved this section. It is now an introduction to chapter three. Please, see green text at lines 108-112.

Comment 3:

2.2 Ethical considerations

129-132 This statement seems a bit casual: since research is conducted inductively some personal or emotional topics might emerge in focus groups. Focus groups should be understood as a very sensitive way of collecting qualitative data, where it is even harder to protect people’s anonymity, since other participants might talk about the group and about other participants.

Response to comment 3:

We agree that focus group interviews can make it more difficult to protect people’s anonymity and that it might be more difficult to express opinions in a group interview. What we attempted to describe was that the only data collected from participants was their views expressed in the focus group interviews which focused on the participants’ views on technology-based support for PA. Hence, the study did not aim at collecting data which can be classified as sensitive personal information (according to the definition in the act). We have clarified this further, please see green text on lines 134-136 We have also rearranged the text a bit to enhance readability.

Comment 4:

2.4 Data Collection

(a) 170: It would be nice to know more on background and experience on focus group moderator.

Response:

(a) More information about the focus group moderator is now given in paragraph 2.4., lines 210-211. ACJ is a physiotherapist with wide clinical experience (more than 30 years), including caring of seniors, and also a researcher experienced as a qualitative interviewer.

(b) 176: Rather use the term “group interviews” than “interviews”.

Response:

The term “group interviews” has been used instead of “interviews”. Please, see green text, line 217.

Comment 5:

2.5.1 Description of the Participants

In my opinion, this sentence/statement could be merged into chapter 2.5.2

Response:

We agree and have merged the statement. Please see line 223 in 2.5.1. (previously 2.5.2).

Comment 6:

Results

(a) 3.2. This chapter should be patched together with the methods chapter, it seems misplaced and redundant here.

Response:

We believe that the referee means 3.1 and agree that this text is redundant with the methods chapter. Therefore, we have removed this section. The text in 3.1 now only contains presentation of results (please see line 263 and below).

(b) Helpful facilitator

300-307 Some more background information on this code “would be useful. How precisely did they express that they needed to feel technology contributing to their well-being? Do they also “blame” themselves for needing technology to give them a push?

Response:

The seniors expressed that they thought that the technology becomes interesting and attractive when the user feels that technology makes her/him feel good. We believe that this is described in the current text (lines 307-313). The participants did nor express that they blamed themselves for needing technology, this was not covered by the interview questions.

(c) Conscious-raising

332-334: Did participants elaborate on these “positive feelings”? 

Response:

The seniors expressed that the technology should be associated with them feeling good, they did not elaborate on different feelings.

(d) 3.2.2. HCPs’ views on needs and requirements on technology-based motivation support for Past

Instead of themes, categories and sub-categories I would suggest a different hierarchy: Categories, sub-categories, dimensions. It was helpful to see that you included a table where coding system is presented.

Response:

We are grateful for the suggestion on an alternative hierarchy. After consideration, we have decided to use the term dimensions in the description of the analysis of differences between seniors’ and HCP’s views (please green text on lines 248-259 in section 2.5.2 and on lines 490, 496 and 527 in section 3.2). We believe that “dimensions” is a better label than theme in this case. However, we have decided to keep the existing hierarchy (themes, categories and sub-categories) in the analysis of each focus group interview. Here, we believe that theme is suitable and more in line with our method reference (Graneheim).

(e) Generally, results of the qualitative data need to be presented in a less descriptive way. Topics/categories/sub-categories should first be presented (as in table) in a self explanatory way. Participant’s quotes can definitely be added for a fuller understanding, but should be used more carefully all over the results part. Moreover, it is not clear if these quotes reflect just a single opinion of a participant or if they are representing all participants as a form of an anchor quote. 

Response:

Initially in the writing process, the topics/categories/sub-categories were presented as tables in the beginning of the results. In order to facilitate reading of the text, the tables were moved to appendices. In order to increase the visibility of the appendices, we have increased the number of references to the appendices in the text. We are happy to include the result tables in the manuscript if you and editor believe that would enhance the manuscript further. 

Based on your suggestion, we have also gone through the text and removed some of the participants’ quotes.

However, we have made a qualitative analysis aiming at describing range and variation in views according to the method described by Graneheim. Therefore, we have not considered whether views have been expressed by one or several participants. Hence, anchor quotes have not been analyzed.

(f) 415-423, I would suggest to better connect different aspects of the categories/sub-categories. How exactly do arguments differ from one another? Paragraph should be re-written stylistically as well, i.g. you often use terms such as “moreover”, “in addition”.

Response:

Section 3.1 has been revised and modified: Several of the quotes from the participants that have been removed and paragraphs have been rewritten in order to better reflect how arguments differ and to improve the text stylistically. Use of terms “moreover” and “in addition” has been decreased.

(g) 506: There was only one part of the quote italic, what does this mean?

Response:

The italic within the quote meant that this was a quote within the quote. This has been replaced with citation marks. Please see green text on lines 485-486. All quotes have been formatted into italics.

Comment 7:

(a) 3.3. Analysis of similarities and differences between seniors’ and HCPs views

550-553: This section seems to belong into the method’s chapter.

Response:

We believe that the referee means section 3.2 (Analysis of similarities and differences between seniors’ and HCPs views) and agree that the mentioned rows are redundant with the methods section. Therefore, we have removed these rows from 3.2. Please see shortened version of the text from 3.2 on lines 490-493.

(b) Increase the senior’s direct motivation for PA

404-407: This quotation could be shortened in order to reach a “thicker description”.

Response:

The quotation has been shortened, please see lines 390-392.

(c) Increase the senior’s indirect motivation for PA through decreased inactivity

434-451: This whole chapter could be merged with the chapter on “increase the senior’s direct motivation for PA”. A “thicker description” in the way of Clifford Geertz would help here to present both a direct and indirect motivation for PA.

Response:

We have removed the word indirect in the title of the section (please see line 409). However, we have decided to keep the two separate chapters since our aim with having them was to distinguish the two different behaviors (activity vs inactivity) from each other. As we see it, there is a difference between focusing on increasing activity and focusing on decreasing inactivity. Focusing on the latter could be a more fruitful approach for motivating inactive persons to increase PA, this was emphasized by the HCPs and therefore described in a separate chapter. 

(d) Support the HCPs’ clinical work

Generally, please shorten this chapter and present the narratives more to the point.

522-530: Where does this quote end?

Response:

The chapter has been shortened and the quote removed (see response to comment 6f). New version is found on lines 45-473.

Comment 8:

Discussion

(a) 610-626: It is very important that you bring theory into the discussion and connect it to your data. Technology should make PA more enjoyable, therefore senior’s intrinsic interest for PA should be investigated and are important, but are senior’s intrinsic motives the only way to make PA more enjoyable? Later you are also stating that “social interaction” is important for seniors, but there’s missing substantial social theory to this point. It is necessary that you include a theory on social interaction and connect it to categories in your data.

Response:

We agree that the paper is improved by relating to theory on social interaction. Our opinion is that this should be done both by clarifying how the results on social interaction relates to SDT (key construct relatedness) and by mentioning additional theory for social interaction. Therefore, we have related the results regarding social interaction to social cognitive theory (please see green text on lines 558-560) and added a point in the discussion section about future studies where we mention to social cognitive theory and health belief model (please, see green text on lines 684-687). In addition, we have clarified how the social interaction contributes to satisfaction of the psychological need of relatedness which is a key construct in SDT (please see green text on rows 555-557). 

(b) 728-774: Maybe an under chapter on the strengths and limitations of the study would help to structure the text?

Response:

We agree that the structure of the discussion could be improved by distinguishing strengths and limitations from the other parts in the discussion, these sections have therefore been moved to the last part of the discussion chapter.

Comment 9:

Conclusion

776: Conclusion could start with 777.

Response to comment 8:

We have removed the sentence which was previously on line 776. Now the conclusion only contains the actual conclusions. Please see line 726.

Round 2

Reviewer 3 Report

Methods, 2.1, Study design

Thank you for substantially making changemnets in the methods chapter.

Further below (251-252), you now mention Graneheim & Lundman's method of analyzing qualitativ data. It would be helpful to already mention / quickly introduce their method in the "study design".

Ethical considerations, 2.2

Thank you for further elaborating your "ethical considerations". Generally, focus groups should be understood as a very sensitive way of collecting qualitative data. Maybe you could just make allowance to this fact by simply stating it....i.g. start from 140 with something like this: "Even though focus groups are a sensitive way of collecting data, the study was performed according…."

Results

265: "two main themes" instead of "two themes"...

Discussion:

556-560: Maybe you could rearrange this part and start with the literature: "Social support has been found to improve the effectiveness of PA interventions". Concretely, social interactions such as XY could contribute to the satisfaction…." This is in accordance with the social cognitive theory that emphasizes the importance of social interaction and model learning for behavioral change

560 [35]. S

Even though I like the idea of connecting your data to the social cognitive theory, it sholud be discussed more critically...does it really lead to behavioral change in every case…?

673: "designing" instead of "design"

Limitations, 709: A qualitative study, especially with two Focus groups shouldn't have the goal to be "representative". Could you please change the wording here?

718-721: In my opinion, interpretation is part of conducting qualitative analysis and I think it's good that you critically mention this procedure. Could you maybe also connect this part with a reference on "interpreting qualitative data"? This would strenghten your argument here substantially.

721-724: This statement seems a bit casual. I would rather close the chapter by the part before on interpretating qualitative data.

5. Conclusions:

Some part of the narration seems a bit redundant here, especially from 735 on....this more belongs to the "discussion". What is the main conclusion here?

Author Response

RESPONSE TO COMMENTS FROM REFEREE

General response to the referee:

Once again, thank you very much for your constructive feedback. It has been very helpful in our work to enhance the paper. We believe that your comments on the qualitative analysis has significantly contributed to the enhancement of the manuscript. The manuscript was proof read by a colleague in the first revision round. If you see need for further improvements of the English language, we are willing to send the manuscript to English language editing from MDPI.

Comment 1:

Methods, 2.1, Study design

Thank you for substantially making changemnets in the methods chapter.

Further below (251-252), you now mention Graneheim & Lundman's method of analyzing qualitativ data. It would be helpful to already mention / quickly introduce their method in the "study design".

Response:

We have mentioned Graneheim & Lundman's method in 2.1 Study design. Please see yellow text on line 120. 

Comment 2:

Ethical considerations, 2.2

Thank you for further elaborating your "ethical considerations". Generally, focus groups should be understood as a very sensitive way of collecting qualitative data. Maybe you could just make allowance to this fact by simply stating it....i.g. start from 140 with something like this: "Even though focus groups are a sensitive way of collecting data, the study was performed according…."

Response:

We have included a phrase about focus group and personal integrity at yellow text on lines 139-140. 

Comment 3:

Results

265: "two main themes" instead of "two themes"...

Response:

The suggested change has been implemented in the text. Please see yellow text on line 268. 

Comment 4:

Discussion:

556-560: Maybe you could rearrange this part and start with the literature: "Social support has been found to improve the effectiveness of PA interventions". Concretely, social interactions such as XY could contribute to the satisfaction…." This is in accordance with the social cognitive theory that emphasizes the importance of social interaction and model learning for behavioral change

560 [35]. S

Response:

We have rearranged the text according to your suggestions, please see yellow text on lines 557-565.

Comment 5:

Even though I like the idea of connecting your data to the social cognitive theory, it sholud be discussed more critically...does it really lead to behavioral change in every case…?

Response:

We have included this in the discussion about social cognitive theory, please see lines 562-562.

Comment 6:

673: "designing" instead of "design"

Response:

The suggested change has been implemented in the text. Please see yellow text on line 679. 

Comment 7:

Limitations, 709: A qualitative study, especially with two Focus groups shouldn't have the goal to be "representative". Could you please change the wording here?

Response:

We have removed representative. Please see yellow text on line 716 (“the results of the study might not be transferable to inactive seniors”).

Comment 8:

718-721: In my opinion, interpretation is part of conducting qualitative analysis and I think it's good that you critically mention this procedure. Could you maybe also connect this part with a reference on "interpreting qualitative data"? This would strenghten your argument here substantially.

Response:

We have connected the text with the reference Silverman, D. Interpreting qualitative data: Methods for Analysing Talk, Text and Interaction, Second ed.; SAGE Publications: London, 2001. Please see yellow text on line 727-728.

Comment 9:

721-724: This statement seems a bit casual. I would rather close the chapter by the part before on interpretating qualitative data.

Response:

We have removed the statement (“However, we believe that the search for similarities and differences through a second analysis according to the principles of inductive content analysis is an interesting approach for comparing qualitative data of different sources.”) as suggested.

Comment 10:

5. Conclusions:

Some part of the narration seems a bit redundant here, especially from 735 on....this more belongs to the "discussion". What is the main conclusion here

Response:

We have clarified the conclusion here. Please see yellow text on lines 740-743.